# Optimal low-depth quantum signal-processing phase estimation

Yulong Dong [1,2] ✉, Jonathan A. Gross [1] & Murphy Yuezhen Niu[1,3] ✉

Quantum effects like entanglement and coherent amplification can be used to drastically enhance the accuracy of quantum parameter estimation beyond classical limits. However, challenges such as decoherence and time-dependent errors hinder Heisenberg-limited amplification. We introduce Quantum Signal-Processing Phase Estimation algorithms that are robust against these challenges and achieve optimal performance as dictated by the Cramér-Rao bound. These algorithms use quantum signal transformation to decouple interdependent phase parameters into largely orthogonal ones, ensuring that time-dependent errors in one do not compromise the accuracy of learning the other. Combining provably optimal classical estimation with near-optimal quantum circuit design, our approach achieves a standard deviation accuracy of $10^{-4}$ radians for estimating unwanted swap angles in superconducting two-qubit experiments, using low-depth ( < 10) circuits. This represents up to two orders of magnitude improvement over existing methods. Theoretically and numerically, we demonstrate the optimality of our algorithm against time-dependent phase errors, observing that the variance of the time-sensitive parameter $\varphi$ scales faster than the asymptotic Heisenberg scaling in the small-depth regime. Our results are rigorously validated against the quantum Fisher information, confirming our protocol's ability to achieve unmatched precision for two-qubit gate learning.

Quantum metrology's efficiency is fundamentally influenced by two critical factors: the Heisenberg limit, which defines how accuracy scales with quantum resources, and the coefficients of this scaling. While a variety of quantum metrology strategies[1–3] successfully adhere to the Heisenberg scaling, the real challenge lies in achieving or even addressing optimality in the scaling coefficients with realistic constraints. This aspect is particularly vital for applications in quantum error correction, where achieving fault-tolerant thresholds demands exceptionally high accuracy in quantum gate characterization. The necessity for deep circuitry, a significant hurdle in practical applications, stems directly from the lack of optimality in these scaling coefficients. This inefficiency is compounded by the challenges of finite coherence times and the amplification of drift errors from low-

frequency noise or control fluctuations. Therefore, current quantum metrology protocols, limited to accuracy levels between $10^{-2}$ and $10^{-3}$ radians[2,3] for estimating gate angles, often fall short of the accuracy ( ~ $10^{-4}$) needed to verify the crossing of fault-tolerant error threshold for quantum error correction and other near-term quantum applications.

In quantum metrology for gate calibration, two primary approaches are used: robust phase estimation (RPE) and randomized benchmarking. RPE, along with its extensions like Floquet calibration, can achieve the Heisenberg limit under ideal conditions and are robust against state preparation and measurement (SPAM) errors across both single and multi-qubit gates[1–3]. However, its practical implementation is limited by the need for deep circuits and resource-intensive, iterative

[1]Google Quantum AI, Venice, California, CA 90291, USA. [2]Department of Mathematics, University of California, Berkeley, Berkeley, CA 94720, USA. [3]Department of Computer Science, University of California, Santa Barbara, Santa Barbara, CA 93106, USA. ✉e-mail: dongyl@berkeley.edu; murphyniu@ucsb.edu

black-box optimizations to ensure accurate calibration. Moreover, its precision drops to between $10^{-1}$ and $10^{-2}$ radians when dealing with the time-dependent drifts common in superconducting qubit systems. Though recent progress[4,5] refine the multiplicative overhead of RPE cost, they focus on the asymptotic regime rather than physically short-depth and noise-robust implementation. Meanwhile, the randomized benchmarking approach, although general, forgoes Heisenberg scaling. It also requires extensive circuit depth to accurately estimate parameters and lacks sensitivity to coherent rotation errors[6–9]. As a result, these prevalent quantum metrology techniques have not yet achieved optimal performance in practice for learning two-qubit gates.

The effectiveness of a quantum metrology scheme can be assessed by the fundamental limits set by both classical and quantum Cramér-Rao bounds[10,11]. To meet the classical Cramér-Rao bounds, the inference subroutines that process measurements to estimate quantum gate parameters must be optimal. Similarly, to achieve the quantum Cramér-Rao bounds, the quantum measurement schemes, characterized by any Positive-Operator-Valued Measurements, must also be optimal[11]. Realizing optimality in both aspects requires refining classical post-processing techniques and the quantum circuit designs used in quantum gate calibration. In this work, we show that the RPE-based multi-parameter phase estimation method requires an additional phase-matching condition: the diagonal elements of the gates must share the same phase. If this condition is violated, the RPE-based method will fail to achieve both Heisenberg scaling and the classical Cramér-Rao bounds when there's more than one phase to learn, even in the absence of quantum noise.

In this work, we propose a metrology protocol that is, by design, robust against realistic time-dependent errors and only requires shallow ( < 10) circuits to achieve up to two orders of magnitudes of improvement over existing methods in the precision of gate-parameter estimates.

## Results

We harness the analytical structure of a class of quantum-metrology circuits using a theoretical toolbox from classical signal processing[12,13], Quantum Signal Processing (QSP)[14–16] and polynomial analysis[17]. QSP allows us to treat the inherent quantum dynamics as input quantum signals and perform universal transformations on the input to realize targeted quantum dynamics as output. Classical signal processing provides methods for analyzing these transformed signals to produce robust estimations. We propose a general gate model, which we term the $U$-gate model, that encapsulates two-level invariant subspace structure in the native gate sets of superconducting, neutral atoms, and ion trap quantum computers. We parameterize the subspace of interest in our model $U$-gates with a set of angle parameters, and provide a metrology algorithm for learning the swap angle $\theta$ and the phase difference $\varphi$.

Our metrology algorithm, which we term Quantum Signal-Processing Phase Estimation (QSPE), separates the estimation of the parameter-free from time-dependent errors ($\theta$) from that which is affected by time-dependent drift ($\varphi$). Interestingly, the parameter $\varphi$ variance shrinks faster than Heisenberg scaling concerning circuit depth in the pre-asymptotic low-depth regime of experimental interest. We analyze the stability of our protocol in the presence of realistic experimental noise and sampling errors. We prove that our method achieves the Cramér-Rao lower bound in the presence of sampling errors and achieves up to $10^{-4}$ STD accuracy in learning swap angle $\theta$ in both simulation and experimental deployments on superconducting qubits. We provide the evaluation of the metrology protocol's quantum Fisher information (QFI) and show that our approach is a factor of two above the quantum CRLB (QCRLB). Furthermore, we demonstrate an interesting transition of the optimal metrology variance scaling as a function of circuit depth $d$ from the pre-asymptotic regime $d \ll 1/\theta$ to the Heisenberg limit $d \to \infty$.

We summarize the main results of our metrology algorithm and start by defining the metrology problem, the learning of a general $U$-gate, followed by an analysis of the QSP circuit with $U$-gates used in our algorithm. Building upon these closed-form results, we propose a phase estimation method combining Fourier analysis with QSP to separate the two gate parameters of interest in their functional forms. Our estimation algorithm enables fast and deterministic data post-processing using only direct linear algebra operations rather than iterative black-box optimizations used in multi-parameter robust phase estimation[1,2]. Moreover, separating the inference of $\theta$ and $\varphi$ enhances the robustness of the phase estimation method against time-dependent errors that predominantly affect the gate parameter $\varphi$, which arise from the time-dependent qubit frequency noise[18–20]. The analysis and modeling of Monte Carlo sampling error also indicate that our phase estimation method achieves the fundamental quantum metrology optimality in a practical regime against realistic errors for near-term devices. We also provide a comprehensive mathematical analysis of methods based on robust phase estimation[2,3], and prove that the vulnerability of phase angle $\varphi$ to time-dependent errors ultimately renders the estimation accuracy of the swap angle $\theta$ exponentially worse than the Heisenberg limit. In addition, we proposed a noise-robust QSPE protocol that enables the estimation of gate parameters even when the gate angles fall outside the confidence regime for phase estimation. Furthermore, we demonstrated QSPE experimentally on 34 superconducting qubits using the Google Quantum AI team's hardware, achieving $10^{-4}$ two-qubit phase estimation accuracy in practice, which improves by two orders of magnitude over standard previous methods. Lastly, we include an empirical noise-robust QSPE protocol that enables the estimation of gate parameters even when the gate angles fall outside of the approximation regime for phase estimation.

### General gate model with two-level system invariant subspace

Our QSPE technique applies to any gate that contains a two-level invariant subspace $\mathcal{B}$, such that states within $\mathcal{B}$ remain within $\mathcal{B}$ when acted upon. Here, we define a general two-level unitary model, which we term the $U$-gate model, around which we base our framework. We parameterize this model gate when restricted to the subspace $\mathcal{B}$ as:

$$[U(\theta, \varphi, \chi, \psi)]_{\mathcal{B}} = \begin{pmatrix} e^{-i\varphi - i\psi}\cos\theta & -ie^{i\chi - i\psi}\sin\theta \\ -ie^{-i\chi - i\psi}\sin\theta & e^{i\varphi - i\psi}\cos\theta \end{pmatrix} \quad (1)$$

We refer to $\theta$ as the swap angle, $\varphi$ as the phase difference, $\chi$ as the phase accumulation during the swap and $\psi$ as the global phase present in the entire gate. Note that $\chi$ cannot be amplified on the same basis as $\theta$ and $\varphi$.

We emphasize that many quantum gates (including single- and multi-qubit gates) can be reduced to the $U$-gate model and thus be characterized by QSPE. For example, in our experimental deployment on the Google Quantum AI superconducting qubits, we study Fermionic Simulation Gates (FsimGates), which are native to superconducting qubit computers. We also remark that not all parameters in our model are necessary for every gate.

The problem of calibrating $U$-gate is to estimate $\theta$, $\varphi$, and $\chi$ for some invariant subspace of $U$ against realistic noise given access to the $U$-gate and basic quantum operations, which we now formalize:

**Problem 1.** (Calibrating $U$-gate). Given access to an unknown $U$-gate, basic quantum gates and projective measurements, how to estimate gate parameters with bounded error and finite measurement samples?

Previous metrology methods[1–3] based on optimal measurements[11] for achieving the Heisenberg limit fall short of providing sufficient accuracy in $\theta$ when $\theta \ll 1$. Two significant factors limit these traditionally regarded "optimal" metrology schemes. First, the accuracy in $\theta$ depends on the amplification factor, i.e., maximum circuit depth. The

relatively low qubit coherence times of superconducting qubits render randomization-based quantum gate learning techniques[6,7,9] impractical due to their inefficient circuit depths. The finite low qubit coherence times[9] of superconducting qubits render randomization-based quantum gate learning techniques[6,7,9] impractical due to their inefficient circuit depths needed to achieve the desired accuracy close to the surface code threshold[21]. Techniques based on robust phase estimation can require prohibitive depths to achieve a meaningful full signal-to-noise ratio for small $\theta$ and require iterative black-box optimizations for their estimators[2] instead of fast, deterministic post-processing for single-parameter phase estimation[1]. Second, time-dependent unitary error in $\varphi$ is prevalent in architectures like Google's superconducting quantum computers[22], which invalidates basic assumptions in traditionally optimal and Heisenberg-limit-achieving metrology schemes.

## Quantum signal-processing phase estimation (QSPE)

In this work, we provide a low-depth phase estimation method for estimating the angles in some invariant subspace of an unknown $U$-gate when the swap angle is small, of order below $10^{-3}$, while facing realistic time-dependent phase errors in $\varphi$. The phase estimation method leverages the structure of periodic circuits analyzed by classical and quantum signal processing and provides a framework to engineer quantum metrology from the perspective of universal quantum signal transformation. We call this type of metrology method Quantum Signal-Processing Phase Estimation (QSPE). Let $\omega$ be a tunable phase parameter and $\{|0_\ell\rangle, |1_\ell\rangle\}$ be the logical basis of the two-level space of interest. Then, QSPE measures the transition probabilities of the quantum circuits corresponding to logical Bell states $|+_\ell\rangle := \frac{1}{\sqrt{2}}(|0_\ell\rangle + |1_\ell\rangle)$ and $|i_\ell\rangle := \frac{1}{\sqrt{2}}(|0_\ell\rangle + i|1_\ell\rangle)$. The transition is measured with respect to the logical basis state $|0_\ell\rangle$. We depict the quantum circuit for QSPE in Fig. 1 with an exemplified two-qubit $U$-gate for simplicity. In the example, the two-level subspace is set to the single-excitation subspace with basis $|0_\ell\rangle = |01\rangle$ and $|1_\ell\rangle = |10\rangle$. Then, the logical Bell state coincides with the conventional Bell state. We remark that the quantum circuit for QSPE can be generalized to multiqubit cases following the recipe outlined in this paragraph. Details and the quantum circuit for QSPE in a general setup are provided in Supplementary Note 2. The transition probability corresponding to the logical Bell state $|+_\ell\rangle$ is denoted as $p_X(\omega; \theta, \varphi, \chi)$, and that corresponding to the logical Bell state $|i_\ell\rangle$ is denoted as $p_Y(\omega; \theta, \varphi, \chi)$.

The measurement probabilities can be viewed as the expectation values of the logical Pauli operators:

$$\langle X_\ell\rangle(\omega; \theta, \varphi, \chi) = 2p_X(\omega; \theta, \varphi, \chi) - 1, \qquad (2)$$

$$\langle Y_\ell\rangle(\omega; \theta, \varphi, \chi) = 2p_Y(\omega; \theta, \varphi, \chi) - 1, \qquad (3)$$

$$\mathfrak{h}(\omega; \theta, \varphi, \chi) = \left\langle \frac{1}{2}(X_\ell + iY_\ell) \right\rangle(\omega; \theta, \varphi, \chi) = \langle a_\ell\rangle(\omega; \theta, \varphi, \chi). \qquad (4)$$

The physical meaning of the reconstructed function $\mathfrak{h}(\omega; \theta, \varphi, \chi)$ coincides with the expected value of the logical annihilation operator which gauges the magnitude of the coherent rotation error in the single-excitation subspace. This observation qualitatively justifies the potential of the candidate function in the proposed phase estimation method.

As outlined in Supplementary Note 3, the reconstructed function derived from measurement probabilities admits an approximated expansion $\mathfrak{h}(\omega; \theta, \varphi, \chi) = \sum_{-d+1}^{d-1} c_k e^{2ik\omega}$ and when $d\theta \le 1$, the coefficients are

$$c_k \approx i\theta e^{-i\chi} e^{-i(2k+1)\varphi} \text{ with } k = 0, \cdots, d-1. \qquad (5)$$

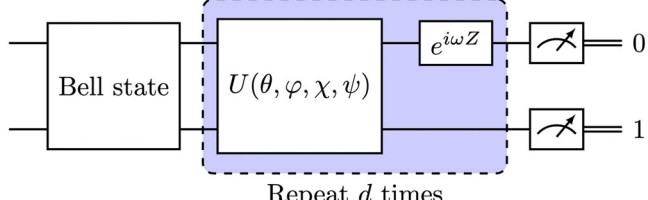

**Fig. 1 | Quantum circuit for QSPE with an exemplified two-qubit $U$-gate.** The input quantum state is prepared to be Bell state in either $|+_\ell\rangle$ or $|i_\ell\rangle$ according to the type of experiment. The quantum circuit enjoys a periodic structure of the unknown $U$-gate and a tunable $Z$ rotation.

Due to Fourier expansion, sampling the reconstructed function on $(2d-1)$ distinct $\omega$ points is sufficient to characterize its information completely. For efficient processing with the Fast Fourier Transformation (FFT), we choose a set of $\omega$ points that are equally spaced. This choice of equally-spaced sampling points not only ensures numerical stability, as demonstrated in textbooks on numerical analysis[23], but also simplifies error analysis, as described in Supplementary Note 4. The second important consequence of this result is that the dependencies on $\theta$ and $\varphi$ are completely separated into the amplitude and the phase of the Fourier coefficients, respectively. The estimation problems of $\theta$ and $\varphi$ are then reduced to two independent linear regression problems. As $\chi$ is not considered, we apply a sequential phase difference to distill the angle $\varphi$:

$$\boldsymbol{\Delta} = (\Delta_0, \cdots, \Delta_{d-2})^\top, \Delta_k := \text{phase}(c_k \overline{c_{k+1}}) = 2\varphi. \qquad (6)$$

Considering the Monte Carlo sampling error due to the finite number of measurements, we derive in Supplementary Note 4 the linear-regression-based estimators of the relevant angles:

$$\hat{\theta} = \frac{1}{d}\sum_{k=0}^{d-1} |c_k| \quad \text{and} \quad \hat{\varphi} = \frac{1}{2}\frac{\mathbf{1}^\top \mathfrak{D}^{-1} \boldsymbol{\Delta}}{\mathbf{1}^\top \mathfrak{D}^{-1} \mathbf{1}}. \qquad (7)$$

Here, $\mathbf{1}$ is an all-one vector and $\mathfrak{D}$ is a $(d-1)$-by-$(d-1)$ constant tridiagonal matrix which coincides with the discrete Laplacian of a central finite difference form (see Definition 15 in Supplementary Note 4 for more details). The structure of $\mathfrak{D}$ comes from differentiating experimental noises when applying sequential phase difference. To obtain $\hat{\chi}$, we defer the task to the metrology circuit in [3, Fig. S5] and do not use QSPE for the task. The main workflow of the QSPE is depicted in Fig. 2 and the Algorithm displayed in Box 1.

## Classical and quantum optimality analysis

The performance of the statistical estimators is measured by their biases and variances. In Supplementary Note 4 B, we derive the performance of QSPE estimators with the following theorem by treating QSPE as linear statistical models. Furthermore, in Supplementary Note 6, we show that QSPE estimators in Eq. (7) are optimal by saturating the Cramér-Rao lower bound (CRLB) of the estimation problem.

**Theorem 1.** In the regime $d \ll 1/\theta$, QSPE estimators in Eq. (7) are unbiased and with variances:

$$\text{Var}(\hat{\theta}) \approx \frac{1}{8d^2 M} \quad \text{and} \quad \text{Var}(\hat{\varphi}) \approx \frac{3}{8d^4\theta^2 M} \qquad (8)$$

where $M$ is the number of measurement shots in each experiment.

We note that the unbiasedness of these estimators holds up to a high-order bias, which is negligible in the target regime. For further details, please refer to Supplementary Note 3.

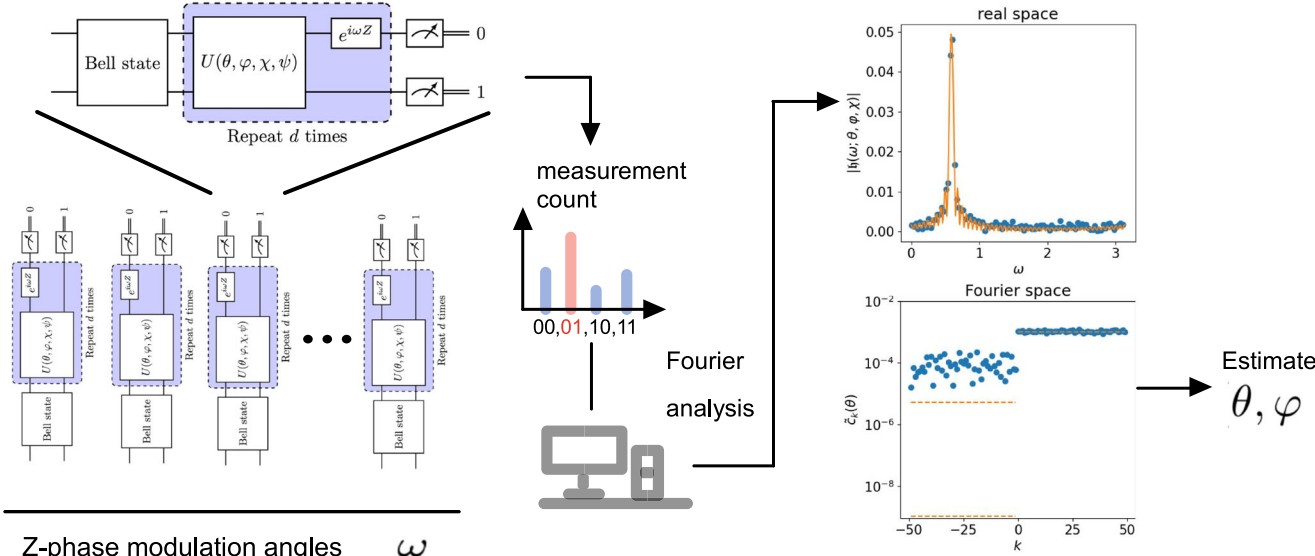

**Fig. 2 | Flowchart of main procedures in QSPE.** The experimental data are collected from depth $d$ quantum circuit experiments featuring equally-spaced phase modulation angles $\omega$, as shown in the left panels. Probabilities from each experiment of different phase modulations are analyzed using Fourier transformation. As illustrated in the right panels, the Fourier-space data are better structured compared to real-space data. Gate angles are then derived using our QSPE estimators.

---

## BOX 1

# Algorithm for inferring unknown angles in *U*-gate with small swap angle using QSPE

**Input:** A *U*-gate $U(\theta, \varphi, \chi, \psi)$, an integer $d$ (the number of applications of the *U*-gate).

Initiate a complex-valued data vector $\mathfrak{h} \in \mathbb{C}^{2\mathbf{d}-1}$.

**for** $j = 0, 1, \cdots, 2d - 2$ **do**

 Set the tunable *Z*-phase angle as $\omega_{\mathbf{j}} = \frac{\mathbf{j}}{2\mathbf{d}-1}\pi$.

 Perform the quantum circuit in Fig. 1 and measure the transition probabilities $p_X(\omega_j)$ and $p_Y(\omega_j)$.

 Set $\mathfrak{h}_{\mathbf{j}} \leftarrow \mathbf{p_X}(\omega_{\mathbf{j}}) - \frac{1}{2} + \mathrm{i}(\mathbf{p_Y}(\omega_{\mathbf{j}}) - \frac{1}{2})$.

**end for**

Compute the Fourier coefficients $\mathbf{c} = \mathrm{FFT}(\mathfrak{h})$.

Compute estimates $\hat{\theta}$ and $\hat{\varphi}$ according to Eq. (7).

**Output:** Estimators $\hat{\theta}, \hat{\varphi}$.

---

**Comparison with Heisenberg limit.** According to the framework developed in ref. 24, the variance of any quantum metrology is lower bound by the Heisenberg limit. It indicates that in our experimental setup when $d$ is large enough, optimal variance scales as $1/(d^3M)$. This seemingly contradicts Theorem 2, where the variance of QSPE $\varphi$-estimator can achieve $1/(d^4M)$. This counterintuitive conclusion is due to the pre-asymptotic regime $d \ll 1/\theta$. In Supplementary Note 6, we analyze the CRLB of QSPE. The optimal variance given by CRLB is exactly solvable in the pre-asymptotic regime $d \ll 1/\theta$.

The key reason behind such faster than Heisenberg limit scaling in pre-asymptotic regime depends on the unique structure of the QSPE circuit: the measurement outcome (Supplementary Equation (7)) concentrates around a constant value regardless of the gate parameter values. Yet when $d$ is large enough to pass to the asymptotic regime, measurement probabilities will take arbitrary values. Furthermore, the

analysis of the CRLB suggests that the optimal asymptotic variance agrees with the Heisenberg limit when $d$ is large enough. This non-trivial transition of optimal variance is theoretically analyzed and numerically justified in Supplementary Note 6. We summarize this nontrivial transition of the optimal variance scaling of QSPE as a phase diagram in Fig. 3 a. To numerically demonstrate the transition, we compute the exact CRLB of QSPE when $\theta = 1 \times 10^{-2}$ and $\theta = 1 \times 10^{-3}$. In Fig. 3 b, the slope of the curve in log-log scale exhibits a clear transition before and after $d = 1/\theta$, which supports the phase diagram in Fig. 3 a. Detailed theoretical and numerical discussions of the transition are carried out in Supplementary Note 6.

**Optimality analysis using Cramér-Rao bounds.** Analyzing Cramér-Rao bounds suggests the optimality of a quantum metrology protocol or the suboptimality leading to further improvement. Given an initialization and measurement, the optimality lies in the analysis of the classical CRLB, which investigates the most information one can retrieve from measurement probabilities. As outlined in Supplementary Note 6 A, the CRLBs are solvable when $d \ll 1/\theta$, which exactly agree with the variance of our estimators derived in Theorem 2. The optimality of our estimator is also validated from the numerical simulation depicted in Fig. 3 b. Such agreement reveals the optimality of our data post-processing. Although linear-regression-based estimators are used, this linearization does not sacrifice the information retrieval in the experimental data. Furthermore, in contrast to other iterative inference methods, our estimators directly estimate angles using basic linear algebra operations, to which stability and fast processing are credited.

Despite the informative indication by analyzing CRLB, it cannot provide direct suggestions on improving initialization and measurement. Such generalization demands the switch to quantum Cramér-Rao lower bound (QCRLB), which requires upper bounding the quantum Fisher information (QFI). As a quantum analog of the classical Fisher information, QFI lies in the center of quantum metrology by providing a fundamental lower bound on the accuracy one can infer from the system of a given resource limit. According to the analysis in ref. 25, the QFI is an upper bound on the Fisher information over all possible measurements. For brevity, we only consider the inference of $\theta$ and hold all other unknown parameters constant in the analysis. However, our analysis can be generalized to the multiple parameter

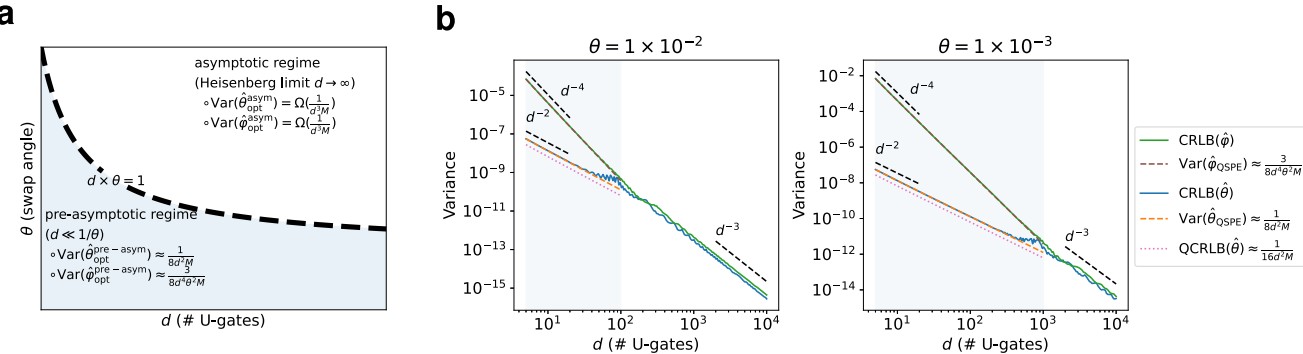

**Fig. 3 | A nontrivial transition of the optimal variance in solving QSPE.** The theoretical analysis of the transition is in Supplementary Note 6. **a** Phase diagram showing the nontrivial transition of the optimal variance in solving QSPE. QSPE estimators attain the optimal variance in the pre-asymptotic regime. **b** Cramér-Rao lower bound (CRLB) and the theoretically derived estimation variance. The single-qubit phases are set to $\varphi = \pi/16$ and $\chi = 5\pi/32$. The number of measurement samples is set to $M = 1 \times 10^5$.

inference by adopting the multi-variable QFI in ref. [25]. In Supplementary Note 6 D, we derive that the average QFI is upper bounded by an integral:

$$\mathfrak{F}_\theta \leq \frac{4}{\pi} \int_0^\pi \frac{\sin^2(d(\omega - \varphi))}{\sin^2((\omega - \varphi))} d\omega = 4d. \tag{9}$$

Here, the integrand gauges the information contained in an experiment with angle $\omega$. The integral stands for the use of equally spaced $\omega$ samples due to the absence of accurate information of $\varphi$. It is worth noting that the integrand is sharply peaked at $4d^2$ when $\omega = \varphi$ which is also referred to as the phase-matching condition. The missing information of $\varphi$ downgrades the average QFI from $4d^2$ to $4d$. However, as discussed in the following subsection, the lack of information about $\varphi$ in existing methods can potentially significantly degrade the Fisher information to $\mathcal{O}(\log(d))$, thereby severely impeding the achievement of Heisenberg-limit scaling in estimation accuracy. Furthermore, this also suggests a finer estimation of $\theta$ when some rough information of $\varphi$ is provided either as a priori or from some preliminary estimation. This improvement is discussed in Supplementary Note 4 C. Consequently, the QCRLB of the QSPE formalism is

$$\text{Var}(\hat{\theta}) \geq \text{QCRLB} \geq \frac{1}{16d^2M}. \tag{10}$$

Compared with Theorem 2, we see differentiation in a constant suboptimal factor of 2, which is explainable. Note that we use two logical Bell states to perform experiments. The advantage is the experimental probabilities of these two experiments form a conjugate pair to reconstruct a complex function for the ease of analysis. This complex function and its properties (see Theorems 7 and 9 in Supplementary Note 3) eventually lead to a simple robust statistical estimator requiring only light computation. In contrast, the data generated from the initialization of one Bell state still contains full information on the parameters to be estimated. However, the highly nonlinear and oscillatory dependency renders the practical inference challenging. Hence, the factor of 2 is due to the use of a pair of Bell states. Although the QFI indicates that inference variance can be lower by removing such redundancy in the initialization, the nature of ignoring practical ease makes it hard to achieve.

### Advantage of QSPE over prior arts

The key behind the success of QSPE is the isolation of $\theta$ and $\varphi$ estimations in Fourier space. This enables the robustness of individual angle estimation against the error and noise in another angle. A prior art that is widely used in the gate calibration in Google's superconducting platform is periodic calibration or Floquet calibration[2,3].

Periodic calibration measures the transition probability between tensor product states $|10\rangle$ and $|01\rangle$ of a periodic quantum circuit with $d$ $U$-gates inside. This differs from our QSPE method which initializes Bell states, though the main body of the quantum circuit is the same. To provide an estimation of angles, periodic calibration uses a black-box optimization to minimize the distance between the parametric ansatz and experimentally measured values. Periodic calibration is based on RPE[1–3] and generalizes RPE to the estimation of multiple angles. Though RPE provably saturates the Heisenberg limit, the actual performance of periodic calibration highly relies on the satisfaction of the so-called phase-matching condition, namely, $\omega = \varphi$. Previous experiments suggest that the violation of the condition would lower the estimation accuracy of $\theta$ angle by a few magnitudes[2]. Because the phase angle $\varphi$ is vulnerable to time-dependent drift errors, the uncertainty of $\varphi$ ultimately ruins the estimation accuracy of the swap angle $\theta$ in periodic calibration. In Supplementary Note 9, we provide a comprehensive mathematical analysis of periodic calibration and prove that even without complex error and noise, the violation of phase-matching condition makes the estimation variance of $\theta$ scale as $1/\log_2(d)$. This is exponentially worse than Heisenberg-limit scaling $1/d^2$ when the phase-matching condition is satisfied as depth increases. A formal statement can be found in Theorem 21 in Supplementary Note 9. Moreover, the complex optimization landscape, detailed in Supplementary Note 9 D, renders the estimation using periodic calibration impractical. Though the periodic calibration with phase-matching, i.e. RPE, has higher Fisher information than QSPE, the hardness of satisfying the phase-matching condition due to finite resource and time-dependent drift error renders the ideal high accuracy estimation challenging. In contrast, by averaging over $\omega$ points, our QSPE is more robust against error by separating $\theta$ and $\varphi$ estimation processes. This is also empirically justified using real data derived from quantum devices in Figs. 4 and 5 in later sections.

### Robustness against realistic errors

We incorporate error mitigation against three different types of errors in our quantum-metrology routine, which we discuss separately below.

1. Decoherence. Exploiting the analysis in the Fourier space provides a fruitful structure for mitigating decoherence. To illustrate, we propose a mitigation scheme for the globally depolarizing error in Supplementary Note 7 A. Numerical simulation shows that the scheme can accurately mitigate the depolarizing error and can drastically improve the performance of QSPE estimators.

2. Time-dependent noise. We numerically investigate the robustness of the QSPE estimators against realistic qubit frequency-drift error[20] based on observation from real experiments. We show in Supplementary Note 7 C that the QSPE estimators preserve their accuracy in the presence of this error.

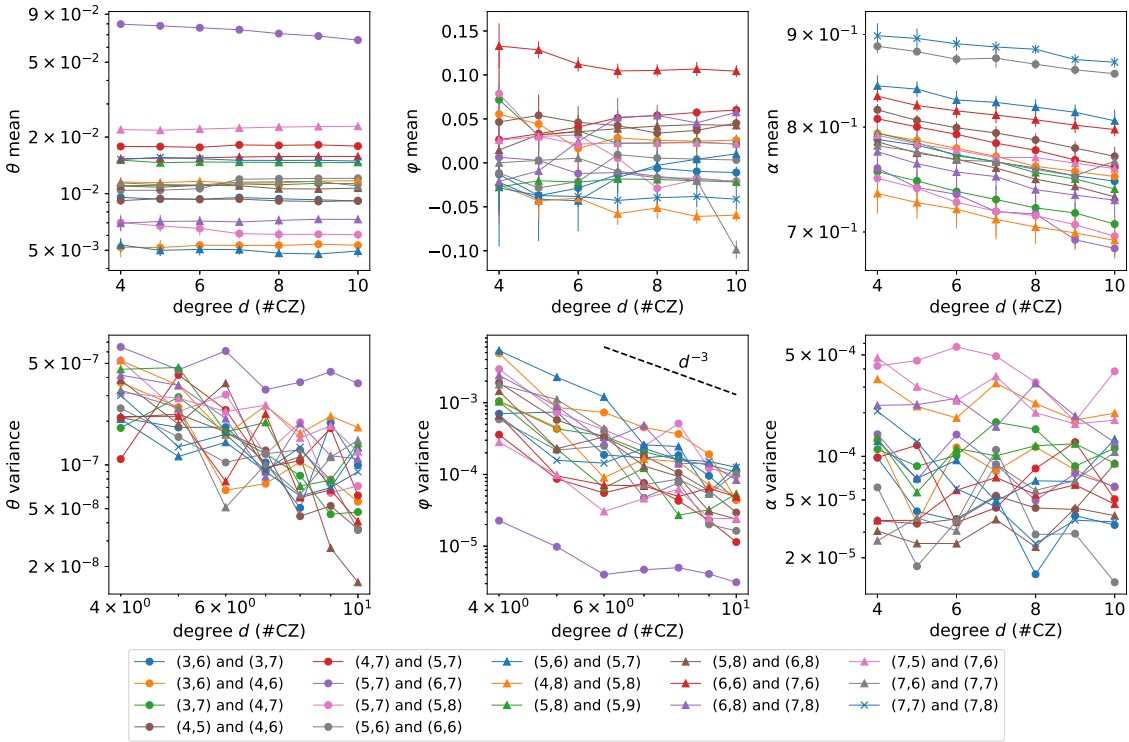

**Fig. 4 | Learning CZ gate with small unwanted swap angle.** Each data point is the average of 10 independent repetitions and the error bars in the top panels stand for the standard deviation across those repetitions. The number of measurement samples is set to $M = 1 \times 10^4$. These columns display the estimated values of gate angles $\theta$, $\varphi$, and circuit fidelity $\alpha$.

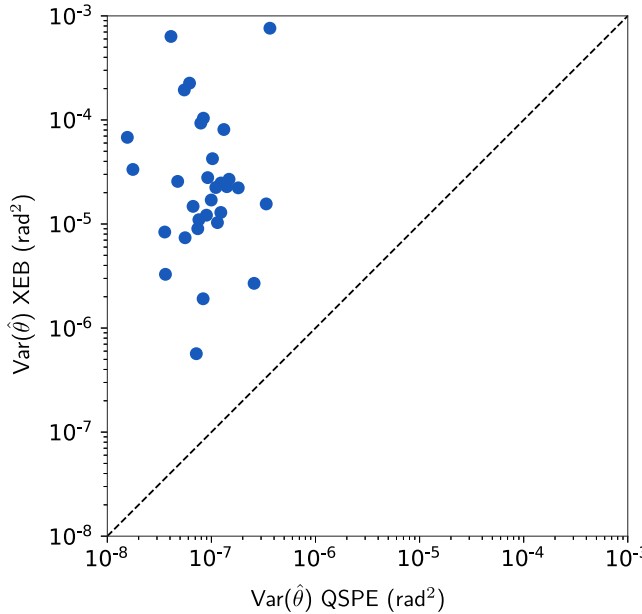

**Fig. 5 |** Comparison of the variance in learning swap angle $\theta$ of CZ gates over seventeen pairs of qubits between QSPE and XEB each repeated for 10 times.

3. Readout errors. In Supplementary Note 7 D, we make an explicit resource estimation for sufficiently accurate mitigation of readout errors.
4. Initial state errors. In Supplementary Note 7 E, we analyze the induced estimation error due to the error in the initial states. We demonstrate that these induced estimation errors are negligibly small in real experimental settings.

We deploy these error mitigation techniques to realize QSPE on a real quantum device. The experimental results are given and discussed in the following section.

## Experimental deployment

In this section, we review the experimental deployment of our metrology method and compare it against the leading alternative methods. We consider learning small swap angles in FsimGates, which are important for fermionic simulation and native to Transmon superconducting qubits. FsimGates are two-qubit $U$-gates. The invariant subspace, referred to as the single-excitation subspace, is spanned by the single-excitation basis $|01\rangle$ and $|10\rangle$. CZ gate is a special Fsim-Gate with zero swap angle. Consequently, FsimGates with small swap angles model the imperfect production of CZ gates whose angle parameter estimation is crucial for applications of CZ gate, including demonstrating surface code[26]. We refer to Supplementary Note 1 C for details.

We use the Google Quantum AI superconducting qubits[9] platform to conduct the experiments described in the Algorithm displayed in Box 1 and Fig. 2. We apply our QSPE method to calibrate $\theta$ and $\varphi$ angles of seventeen pairs of qubits on which CZ gates act. Each CZ gate qubit pair is labeled by the coordinates $(x_1, y_1)$ and $(x_2, y_2)$ of both qubits on a two-dimensional-grid architecture. We plot the statistics of the learned gate angles in Fig. 4: the unwanted swap angle for most qubits are small, of order below $10^{-2}$. In comparison, periodic calibration yields unstable estimates with a highly variant standard deviation across different runs (see Supplementary Fig. 2 in Supplementary Note 1 A), a result of its sensitivity to time-dependent errors.

The performance advantage of QSPE over prior art lies in its robustness against time-dependent noise in the single-qubit phase $\varphi$. In traditional methods, such as XEB and robust phase estimation[2], the measurement observables are nonlinear functions of both $\varphi$ and $\theta$, so if there is time-dependent drift in $\varphi$ during each experiment, or over

different repetitions of the same experiment routine, the value of inferred $\theta$ will be directly affected (see Supplementary Fig. 2 in Supplementary Note 1 A). In comparison, QSPE is tolerant to a realistic time-dependent error in $\varphi$ when estimating the swap angle $\theta$ due to the analytic separation between the two parameters through signal transformation, signal processing and Fourier analysis.

To validate the stability othe f QSPE method, we repeat the same phase estimation routine on each CZ gate pair over 10 independent repetitions. This allows us to bootstrap the variance of the QSPE estimator on $\theta$ and $\varphi$. We show the measured variance and mean of the $\theta$ and $\varphi$ estimates on seventeen CZ-gate pairs over different circuit depths $d$ used in QSPE in Fig. 4. It is important to note that we apply the technique discussed in Supplementary Note 7 A to mitigate globally depolarizing errors using information from Fourier space. This error mitigation procedure estimates a globally depolarizing circuit fidelity $\alpha$ for each pair of qubits on which CZ gates act, as shown in the right-most column of Fig. 4. We observe that the circuit fidelity demonstrates a clear exponential decay with increasing circuit depth, which is consistent with our theoretical analysis[27]. We show that on average the variance in $\theta$ estimates is around $10^{-7}$ for a depth-10 QSPE experiment. This corresponds to $3 \times 10^{-4}$ in STD, which is one to two orders of magnitude lower than the value of $\theta$ itself. In comparison, we also performed the same set of experiments using XEB and compared the results to QSPE in Fig. 5. The variance of $\theta$ inferred by XEB is of order $10^{-4}$ (three orders of magnitude larger than QSPE). Consequently, we show that XEB and periodic calibration are insufficient to learn the value of $\theta$ in our experiments with a larger than unity signal-to-noise ratio.

### Generalization of QSPE for an extended range of swap angles

In earlier sections, we demonstrate the QSPE algorithm's effectiveness for estimating angles when the swap angle is of small magnitude. The actual use of QSPE is not limited to this parameter regime. In this subsection, we propose a generalization of QSPE for general swap angles. Theoretical analyses in Supplementary Note 4 show that noise in Fourier space is significantly reduced, which consequently suggests the algorithm design using Fourier-transformed data. The key observation is that the exact expression of the amplitude of Fourier coefficients, which is referred to as $A_k(\theta)$, can be efficiently solved. Hence, given a set of experimental data, we can estimate the swap angle $\theta$ by aligning experimental amplitudes with theoretical expressions, effectively solving systems of nonlinear equations. When using $d$ $U$-gates per circuit, as discussed in Supplementary Note 5, we outline an empirical noise-robust algorithm estimating $\theta$ to error $\epsilon$ using $\mathcal{O}(d \log(d)\epsilon^{-1})$ classical operations. In the numerical results in Supplementary Note 5, we demonstrate that the $\theta$-estimation error remains below $5 \times 10^{-4}$ for general $\theta$, even with only five $U$-gates. Thanks to Fourier transformation, the angle $\varphi$ is inferred from the phases of Fourier coefficients, decoupling its estimation from $\theta$. It also allows the use of the QSPE $\varphi$-estimator in Eq. (7) for varied swap angles, despite the signal-to-noise ratio varies with different swap angles. This is analyzed in the numerical results in Supplementary Note 5.

### Learning quantum crosstalk with QSPE

An important application of QSPE, thanks to its exceptional sensitivity in measuring small rotations, is in learning quantum crosstalk amplitudes. Quantum crosstalk errors arise from unintended quantum interactions between qubits, which become more problematic when gates operate together. These errors create correlations that are either spatial or temporal, posing a challenge to achieving fault-tolerant quantum computation. In the case of tunable superconducting transmon qubits[28], interactions between two qubits are facilitated by a third "coupler" qubit placed between each pair. This setup allows for the two-qubit interaction to be controlled−turned on or off−by adjusting the coupler qubit's frequency. Yet, even with control over the coupler qubit's frequency, there remains a non-zero

amount of coupling between neighboring qubits' different levels. This coupling mimics the system's Bose-Hubbard coupling Hamiltonian: $H_{\text{crosstalk}} = g_{\text{crosstalk}}\left(\hat{a}_1\hat{a}_2^\dagger + \hat{a}_1^\dagger\hat{a}_2\right)$, where $\hat{a}_i$ denotes the bosonic annihilation operator for the $i$-th qubit.

Furthermore, the main effect of quantum crosstalk in qubit subspace can be described by a rotation within the single qubit subspaces span $\{|10\rangle, |01\rangle\}$ of the two qubits affected by crosstalk. Without any gate operation, the crosstalk's impact over a period $\Delta t$ follows the same pattern as shown in Supplementary Equation (4), where $\theta = g_{\text{crosstalk}}\Delta t$ depends on the crosstalk strength and the duration of crosstalk interaction. This insight allows for the learning of crosstalk effects using QSPE by substituting the $U$-gate in Fig. 1 with an idle gate for an appropriate duration $\Delta t$, ensuring that $g_{\text{crosstalk}}\Delta t$ is sufficiently large to be measurable, yet not so large as to compromise the assumptions underlying QSPE. For example, by setting the circuit depth to $d = 5$ and $\Delta t$ to 200 ns, the precision in measuring $g_{\text{crosstalk}}$ can reach around 10 MHz, markedly surpassing the accuracy of state-of-the-art results in similar systems, which are around 1 MHz[9].

## Discussion

Our proposed QSPE estimators leverage the polynomial structure of periodic circuits through classical and quantum signal processing. These analytics helped us to design an algorithm where the estimation of the swap angle $\theta$ is largely decoupled from that of the single-qubit phases $\varphi$ and $\chi$. When some constant phase drift is imposed on the system, the inference is not affected, thanks to the robustness of the Fourier transformation and sequential phase difference. We demonstrate such robustness against realistic errors, including drift errors in both numerical simulations and deployment on quantum devices. Such robustness is essential in achieving a record level of accuracy not demonstrated before in superconducting qubits. An additional error-mitigation method against globally depolarizing error is further achieved here using the difference in the Fourier coefficients.

Prior to this work, error mitigation routines had been largely separated from quantum metrology protocols, preventing us from achieving the ultimate limit permitted by physics. Our successful combination of error mitigation with metrology hinges upon treating quantum metrology as a type of quantum signal processing: amplifying a given quantum signal while de-amplifying the unwanted experimental noise. Our work, therefore, opens directions for using advanced quantum simulation techniques in the design of quantum metrology algorithms in order to achieve properties necessary for high-accuracy gate learning against realistic environmental noise.

Our future work aims to generalize the optimality of the metrology algorithm against more types of errors and schemes. First, our proof of QSPE estimators' optimality is based on analyzing the Monte-Carlo sampling error. To fully optimize the design of statistical estimators against all types of error in addition to sampling errors requires modeling and studying the behavior and statistics of all types of dominant realistic error using tools from classical statistics, Bayesian inference, and machine learning. Secondly, here we do not optimize all possible state initialization and measurement schemes. Although theoretical analysis and numerical simulation prove that QSPE estimators are optimal in the given parameter regime and the given state preparation and measurement scheme, it remains an open question whether we can derive the optimal estimators in the most generic setting in Problem 1 by optimizing circuit structure at initialization and measurement steps. Thirdly, the QSPE estimators are designed for the low-depth regime ($d < 10$). The restriction to this finite-depth setting is tied in with our main objective of mitigating the detrimental effect of time-dependent noise where deeper circuit depths introduce more drift and decoherence error. A related work employing a dynamical-decoupling-based scheme[29] achieves similar accuracy; however, it is not effective

in the low-depth limit of our algorithm and lacks optimality guarantees due to its reliance on decoupling sequences, necessitating deeper circuits. This low-depth limit can be lifted if we can fully mitigate various noise effects that kick in at deeper depth. Furthermore, generalizing a deterministic estimator from our work to a variational one can also offer greater flexibility and optimality but requires a deeper understanding of the landscape inherited from the QSPE structure. Lastly, qubitization techniques[15,30] and cosine-sine decomposition[31,32] provide powerful two-dimensional subspace representations associated with any unitary matrix. These techniques could potentially generalize our estimation method to large systems with a large number of qubits, which will be our future work.

## Methods

### Quantum experiments
Quantum experiments in our work are conducted using the Google Quantum AI superconducting qubits[9] platform. Our QSPE method is detailed in the algorithm shown in Box 1 and Fig. 2. The details of the XEB method, used in Fig. 5, are discussed in Supplementary Note 1 A.

### Numerical simulations
All numerical tests are implemented in python. The numerical studies in Supplementary Note 4 E and Supplementary Note 7 C are conducted using Cirq. The study of our method's robustness against quantum errors in Supplementary Note 7 C is performed using Cirq's built-in noisy simulator. Other numerical studies, which do not directly involve quantum circuits, are performed using numpy.

## Data availability
All data presented in this work are visualized in the figures and tables within the main text and the Supplementary Information file.

## Code availability
The codes that support the finding are available at[33].

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

## Acknowledgements
This work is partially supported by the NSF Quantum Leap Challenge Institute (QLCI) program through grant number OMA-2016245 (Y.D.). The authors thank discussions with Ryan Babbush, Connor Clayton, Zhiyan Ding, Zhang Jiang, Lin Lin, Shi Jie Samuel Tan, Vadim Smelyanskiy and K. Birgitta Whaley. The authors thank Rajarishi Sinha from Google Cloud AI for his help in open-sourcing the code supporting this finding.

## Author contributions
M.N. conceived the project, carried out some of the theoretical analysis, and led and coordinated the project. Y.D. contributed to the original idea

and carried out the theoretical analysis and numerical simulation to support the study. M.N. and J.G. discussed and carried out experiments on real quantum devices to support the study. All authors contributed to the writing of the manuscript.

## Competing interests

The authors declare no competing interests.
