## [Transparent Peer Review file · Nature Communications]

Optimal Low-Depth Quantum Signal-Processing Phase Estimation

Corresponding Author: Dr Yulong Dong

Version 0:

Reviewer comments:

Reviewer #1

(Remarks to the Author)

Review:

The authors proposed a new framework, QSPE, for estimating the parameters of a quantum gate. This method achieves high accuracy with relatively shallow circuits, which is highly desirable. In the small swap angle setting, crucial for quantum gate calibration in quantum computers, this paper demonstrates that their method is optimal regarding both classical and quantum Cramér-Rao bounds, up to at most a constant factor of 2. Compared with previous approaches, the proposed method is more efficient and robust against realistic errors, as it separates different parameters to be estimated. Both numerical tests and experiments on quantum computers show the advantages of the current method. The authors provided sufficient details and code to ensure the work can be reproduced. I suggest publishing this paper after addressing some minor revisions.

Questions:

1. In the first paragraph, the authors stated that their approach improves the "scaling coefficients." Does this improved scaling coefficient mean the constant factors of the Heisenberg limit scaling, or does it have a richer meaning, such as a more accurate description with respect to parameter d in the pre-asymptotic setting?
2. Do we assume that each implementation of the U gate has the same parameters θ, ϕ, χ, ψ ? My understanding based on your paper is that ϕ can change over time, which is referred to as time-dependent drift, while other parameters do not change. Is this correct? If so, it might be helpful to add some explanation in the paper.
3. Theorem 2 states that Eq.(7) gives an unbiased estimation. However, Eq.(5) implies $|c_k| \approx |\theta|$, which is not an exact equality. Does this contradict the statement of "unbiased"? It seems it could have a bias of order θ^3 .
4. In Figure 4, what is the α in the y-label of the two plots on the right? Why do the plotted quantities differ for qubits in different locations?
5. Some possible typos:
 - Line 302: "alrge" should be "large."
 - Line 579 and 581: Is the meaning of "mHz" and "MHz" the same?

(Remarks on code availability)

Reviewer #2

(Remarks to the Author)

In their manuscript titled „Optimal Low-Depth Quantum Signal-Processing Phase Estimation“ the authors derive and demonstrate a pulse sequence and analysis method to estimate the rotation angles of a quantum operation in a two level

subspace to high accuracy. The authors lay out the analytical derivation of their method in great detail, rigorously working out the boundaries and limits of the parameter estimation.

The parameter estimation works on a generalized U-gate which includes the most common implementations of two-qubit entangling operations. It utilizes a particularly low-depth quantum circuit to estimate both the swap angle as well as the phase rotation independently. It uses a Bell state in the relevant subspace as an input resource and repetitions of the gate in question to amplify the errors in angles of the rotation. By sampling distinct frequencies of a well understood fourier expansion, the method is able to separate the effects of different rotation angles.

Finally, the Authors perform a thorough investigation into the limits of their method and compare it to prior arts of angle estimation methods. They demonstrate the method on the Google Quantum AI hardware, employing several error mitigation methods. They characterize 17 implementations of a CPHASE gate, showing increased accuracy in comparison to standard methods which the authors attribute to time-dependent errors.

In conclusion, the manuscript is written in great detail and many properties of the method are deeply explored. The results are impressive and demonstrated both analytically and experimentally, showing great improvement over the state of the art. The method is discussed in several contexts beyond the initial purpose, such as the extension to arbitrary angle values and the inference of crosstalk between qubits. I would recommend the manuscript for publication, with some minor points of interest to be addressed:

1. The method uses a Bell state in the relevant subspace as resource. Typically, such states are prepared by using the unitary operation under investigation, as quantum systems seldom afford to operate multiple entangling gates. Could the authors comment on the effect of angle mismatches in the preparation of the bell state?
2. Related to point 1, could the method be extended to be used in a different subspace to infer e.g. leakage during a gate?
3. The sampling of the fourier frequencies ω_j is not clearly derived, although given in the algorithm 1. A short statement regarding the reasoning for this choice would help in following the method.
4. In their experimental implementation, the measured values inferred from the algorithm seem to be changing with the parameter d much more than what would be considered within error bars. This suggests some systematic scaling with d . In my understanding, none of the parameters should vary with d . How is this in agreement with an accurate estimation of the parameters?
5. The caption of Figure 2 is too short. I realize the figure is described in the text, but it should be clearly described in the caption.
6. In general, the manuscript lacks in giving citations for some claims. Examples are
 - a. The accuracies in line 24 and 28
 - b. The Heisenberg scaling in line 45
 - c. The inefficient circuit depth of superconducting qubits in line 392
 - d. The previous experiments mentioned in line 407
 - e. I would urge the authors to include a few more sources to those claims and others.
7. There are some spelling errors that I found:
 - a. Starting a section with „Their ...“ reads strange , line 271
 - b. Large , line 302
 - c. Out of place word in line 373
 - d. „pairs of CZ gates“ is very confusing in 474
 - e. The top left plot in figure 4 has only a single y tick, making it hard to interpret
 - f. Missing word in line 545

Overall, I think the manuscript is of great quality and I am happy to recommend it for publication in Nature Communications.

(Remarks on code availability)

Version 1:

Reviewer comments:

Reviewer #1

(Remarks to the Author)

The questions are addressed and the current version of the manuscript looks good to me.

(Remarks on code availability)

Reviewer #2

(Remarks to the Author)

With the applied changes, I am happy to recommend the manuscript for publication!

(Remarks on code availability)

We appreciate the reviewers’ valuable comments and the careful reading of our manuscript. We provide our response to these comments and improve the manuscript accordingly. The changes in the manuscript are highlighted in red color.

Comments by Reviewer #1

1. *In the first paragraph, the authors stated that their approach improves the "scaling coefficients." Does this improved scaling coefficient mean the constant factors of the Heisenberg limit scaling, or does it have a richer meaning, such as a more accurate description with respect to parameter d in the pre-asymptotic setting?*

Response: We thank the reviewer for carefully reading our manuscript. The improved scaling coefficient we have discovered in this protocol has a richer meaning than a mere constant factor of the Heisenberg limit in three ways. First, Heisenberg limit scaling only applies in asymptotic deep circuit depth limit $d \rightarrow \infty$, and it does not concern the optimality of finite depth algorithms. Our method is the first provably optimal metrology method for learning multiple parameters at the same time. Second, we discovered that at the short-depth limit, our algorithm and the optimal Cramér-Rao bound deviate from the traditional Heisenberg limit at the deep depth limit, where some parameters can converge faster in depth than the asymptotic limit thanks to the use of Fourier analysis. Thirdly, our analytical results yield an explicit closed-form estimation performance in the pre-asymptotic regime where the Heisenberg limit does not directly apply and does not have an explicit form beyond scaling.

2. *Do we assume that each implementation of the U gate has the same parameters $\theta, \varphi, \chi, \psi$? My understanding based on your paper is that φ can change over time, which is referred to as time-dependent drift, while other parameters do not change. Is this correct? If so, it might be helpful to add some explanation in the paper.*

Response: The reviewer has correctly pointed out that our error model here, concerning the superconducting qubits from Google’s device, has a specific assumption on the time-dependent error only on φ . In our manuscript, we have added clarification on the time-dependent error model for φ to justify this assumption.

On Page 2, around line 130, in the “Results” section:

“Moreover, separating the inference of θ and φ enhances the robustness of the phase estimation method against time-dependent errors that predominantly affect the gate parameter φ , which arise from qubit frequency noise [MartinisEtAl2003,FoxenEtAl2020,WudarskiEtAl2023].”

Here we clarify that the time-dependent noise only occurs in φ , due to the time-dependent frequency noise of superconducting qubits.

3. *Theorem 2 states that Eq.(7) gives an unbiased estimation. However, Eq.(5) implies $|c_k| \approx |\theta|$, which is not an exact equality. Does this contradict the statement of "unbiased"? It seems it could have a bias of order θ^3 .*

Response: We thank the reviewer for their detailed reading of our manuscript and for raising this question. We acknowledge that the unbiasedness of our estimator is subject to a high-order bias of order θ^3 , as a consequence of the approximation analysis. In our revised manuscript, we have clarified this high-order bias at the first mention of the estimator’s unbiasedness.

Changes to text: Below Theorem 2, we add

We note that the unbiasedness of these estimators holds up to a high-order bias, which is negligible in the target regime. For further details, please refer to Appendix C.

4. In Figure 4, what is the α in the y-label of the two plots on the right? Why do the plotted quantities differ for qubits in different locations?

Response: We appreciate the reviewer highlighting this point. The parameter α represents the globally depolarizing circuit fidelity, defined as $\varrho_{\text{noisy}} = \alpha\varrho_{\text{exact}} + (1 - \alpha)I/4$ in a two-qubit system. This fidelity parameter can be estimated from Fourier-space information using the technique we developed, detailed in Appendix G1. In different experiments, the CZ gates might act on different qubit pairs. Consequently, the circuit fidelities differ in different qubit pairs. Additionally, as the parameter d increases, the circuit fidelity tends to decrease due to increased noise within the system. This metric reflects the quality of the qubits and provides valuable insights for selecting qubits in more complex experimental setups. In our revised manuscript, we have clarified the definition and significance of this parameter both in the main text and in the figure captions.

Changes to text: In the paragraph analyzing numerical results in Figure 4, we add

It is important to note that we apply the technique discussed in Appendix G 1 to mitigate globally depolarizing errors using information from Fourier space. This error mitigation procedure estimates a globally depolarizing circuit fidelity α for each pair of qubits on which CZ gates act, as shown in the right-most column of Figure 4. We observe that the circuit fidelity demonstrates a clear exponential decay with increasing circuit depth, which is consistent with our theoretical analysis [BoixoIsakovSmelyanskiyEtAl2018].

In the caption of Figure 4, we add

These columns display the estimated values of gate angles θ, φ , and circuit fidelity α .

5. Some possible typos:

Response: We thank the reviewer for carefully reading our manuscript. We have corrected the typos noted in the comments, as well as other typos we subsequently identified. These corrections are marked red in the revised manuscript.

- Line 302: "alrge" should be "large."

Changes to text:... when d is large enough...

- Line 579 and 581: Is the meaning of "mHz" and "MHz" the same?

Changes to text:... can reach around 10 MHz...

Comments by Reviewer #2

1. The method uses a Bell state in the relevant subspace as resource. Typically, such states are prepared by using the unitary operation under investigation, as quantum systems seldom afford to operate multiple entangling gates. Could the authors comment on the effect of angle mismatches in the preparation of the bell state?

Response: We appreciate the reviewer's insightful suggestion. In the revised manuscript, we add a subsection analyzing the induced estimation error due to the error in the initial states. We demonstrate that these errors are negligibly small in real experimental settings. Specifically, when the imperfect CZ gate under investigation has a θ value of 10^{-3} and the initial state is prepared using that CZ gate, our results show that the induced estimation error for θ is around 10^{-6} .

Changes to text: In the main text, we add a summarizing discussion in the section titled "Robustness against realistic errors":

Initial state errors. In Appendix G 5, we analyze the induced estimation error due to the error in the initial states. We demonstrate that these induced estimation errors are negligibly small in real experimental settings.

In Appendix G 5, we add a subsection analyzing the effect of initial state errors in both theoretical and numerical ways.

2. *Related to point 1, could the method be extended to be used in a different subspace to infer e.g. leakage during a gate?*

Response: We appreciate the reviewer for highlighting this aspect. When the leakage can be modeled as dynamics within a two-level system, our method is directly applicable to inferring the strength of gate leakage. Specifically, in the section titled “Learning quantum crosstalk with QSPE” on Page 8, we discuss how our approach can be used to learn crosstalk, where information leaks beyond the intended qubit subspace. Exploring the generalization of our method to systems beyond two levels and understanding its potential limitations will be a focus of our future research.

3. *The sampling of the Fourier frequencies ω_j is not clearly derived, although given in the algorithm 1. A short statement regarding the reasoning for this choice would help in following the method.*

Response: We appreciate the reviewer’s valuable suggestion. We have provided further details on the choice of ω points in the main text.

Changes to text: For efficient processing with the Fast Fourier Transformation (FFT), we choose a set of ω points that are equally spaced. This choice of equally-spaced sampling points not only ensures numerical stability, as demonstrated in textbooks on numerical analysis [Trefethen2019SIAM], but also simplifies error analysis, as described in Appendix D.

4. *In their experimental implementation, the measured values inferred from the algorithm seem to be changing with the parameter d much more than what would be considered within error bars. This suggests some systematic scaling with d . In my understanding, none of the parameters should vary with d . How is this in agreement with an accurate estimation of the parameters?*

Response:

We appreciate the reviewer for highlighting this aspect. We would like to note that the numerical results presented in Figure 4 are derived from experiments conducted on experimental superconducting quantum computing platforms, which are subject to realistic quantum errors and noises. The dominant error the reviewer pointed out is a time-dependent error in φ caused by low-frequency noise in qubit frequency for example. As shown in the middle panel of Figure 4, the value of φ varies with different circuit depths and equivalently different experimental runtimes. This variation suggests that this angle is sensitive to time-dependent drift errors, aligning with sensitivities reported in the literature, and our error model assumptions. Despite our effort to include as many sources of errors that are important as there are, experimental uncertainties and finite sample size can still cause deviation from the perfectly unbiased estimator. We acknowledge the reviewer’s concerns regarding potential systematic quantum errors underlying our experiments. As discussed in the “Discussion” section, addressing the modeling of these complex quantum errors in real-time feedback-enabled algorithms will be part of our future work. We would like to clarify that the trend of approximate consistency in the estimated values across our current experiments indicates that our estimation procedure remains robust. Furthermore, our numerical simulations presented in Figures 10 and 11 in the Appendix support that our estimator consistently predicts the true gate angles despite sampling errors and depolarizing quantum errors.

5. The caption of Figure 2 is too short. I realize the figure is described in the text, but it should be clearly described in the caption.

Response: We thank the reviewer for the useful suggestion. We have expanded the caption of Figure 2.

Changes to text: Flowchart of main procedures in QSPE. The experimental data are collected from depth d quantum circuit experiments featuring equally-spaced phase modulation angles ω , as shown in the left panels. Probabilities from each experiment of different phase modulations are analyzed using Fourier transformation. As illustrated in the right panels, the Fourier-space data are better structured compared to real-space data. Gate angles are then derived using our QSPE estimators.

6. In general, the manuscript lacks in given citations for some claims. Examples are

- a. The accuracies in line 24 and 28
- b. The Heisenberg scaling in line 45
- c. The inefficient circuit depth of superconducting qubits in line 392
- d. The previous experiments mentioned in line 407
- e. I would urge the authors to include a few more sources to those claims and others.

Response: We thank the reviewer for the useful suggestion. We add several citations to the places mentioned above.

Changes to text:

- a. ... current quantum metrology protocols, limited to accuracy levels between 10^{-2} and 10^{-3} radians [NeillEtAl2021,AruteEtAl2020]
- b. ... Meanwhile, the randomized benchmarking approach, although general, forgoes Heisenberg scaling [PhysRevA.77.012307,PhysRevA.85.042311]
- c. ... The finite qubit coherence times [GoogleQuantumSupremacy2019] of superconducting qubits render randomization-based quantum gate learning techniques [PhysRevA.77.012307, PhysRevLett.106.180504,GoogleQuantumSupremacy2019] impractical due to their inefficient circuit depths needed to achieve the desired accuracy close to surface code threshold [AcharyaEtAl2024].
- d. ... Previous experiments suggest that the violation of the condition would lower the estimation accuracy of θ angle by a few magnitudes [NeillEtAl2021].

7. There are some spelling errors that I found:

Response: We thank the reviewer for carefully reading our manuscript. We have corrected the typos noted in the comments, as well as other typos we subsequently identified. These corrections are marked red in the revised manuscript.

a. Starting a section with "Their ..." reads strange, line 271

Changes to text: We change the first sentence in that paragraph to: The performance of the statistical estimators is measured by their biases and variances.

b. Large , line 302

Changes to text:... when d is large enough...

c. Out of place word in line 373

Changes to text:... reconstruct a complex function ~~that~~ for **the** ease of analysis ...

d. "pairs of CZ gates" is very confusing in 474

Changes to text:... pairs of **qubits on which** CZ gates **act** ...

e. The top left plot in figure 4 has only a single y tick, making it hard to interpret

Changes to text: We have added additional y-axis ticks to the subfigures in the left column to improve their interpretability.

f. Missing word in line 545

Changes to text: An important **application** of QSPE ...